# Soil Carbon Stocks and Greenhouse Gas Mitigation of Agriculture in the Brazilian Cerrado—A Review

**DOI:** 10.3390/plants12132449

**Published:** 2023-06-26

**Authors:** Arminda Moreira de Carvalho, Douglas Rodrigues de Jesus, Thais Rodrigues de Sousa, Maria Lucrécia Gerosa Ramos, Cícero Célio de Figueiredo, Alexsandra Duarte de Oliveira, Robélio Leandro Marchão, Fabiana Piontekowski Ribeiro, Raíssa de Araujo Dantas, Lurdineide de Araújo Barbosa Borges

**Affiliations:** 1Embrapa Cerrados, BR-020, km 18, Planaltina 73310-970, DF, Brazil; alexsandra.duarte@embrapa.br (A.D.d.O.); robelio.marchao@embrapa.br (R.L.M.); fbn2.ribeiro@gmail.com (F.P.R.); rahdantas08@gmail.com (R.d.A.D.);; 2Faculty of Agronomy and Veterinary Medicine, University of Brasília, Campus Darcy Ribeiro, Brasília 70910-970, DF, Brazil; rodrigues-douglas@outlook.com (D.R.d.J.); lucreciaunb@gmail.com (M.L.G.R.); cicerocf@unb.br (C.C.d.F.)

**Keywords:** agricultural management, agricultural practices, Brazilian Cerrado, greenhouse gas emissions, organic matter, soil organic carbon

## Abstract

New agricultural practices and land-use intensification in the Cerrado biome have affected the soil carbon stocks. A major part of the native vegetation of the Brazilian Cerrado, a tropical savanna-like ecoregion, has been replaced by crops, which has caused changes in the soil carbon (C) stocks. To ensure the sustainability of this intensified agricultural production, actions have been taken to increase soil C stocks and mitigate greenhouse gas emissions. In the last two decades, new agricultural practices have been adopted in the Cerrado region, and their impact on C stocks needs to be better understood. This subject has been addressed in a systematic review of the existing data in the literature, consisting of 63 articles from the Scopus database. Our review showed that the replacement of Cerrado vegetation by crop species decreased the original soil C stocks (depth 0–30 cm) by 73%, with a peak loss of 61.14 Mg ha^−1^. However, when analyzing the 0–100 cm layer, 52.4% of the C stock data were higher under cultivated areas than in native Cerrado soils, with a peak gain of 93.6 Mg ha^−1^. The agricultural practices implemented in the Brazilian Cerrado make low-carbon agriculture in this biome possible.

## 1. Introduction

The Brazilian Cerrado biome, a vast tropical savanna ecoregion in eastern Brazil, covers an area of 204 million hectares. Approximately 111 million hectares (46%) of the native vegetation is still intact. Pastures occupy 25.13% of the soil in the Cerrado (49.874.051 ha) [1], and annual and perennial crops account for about 12.85% of the area (25.510.455 ha) [2]. This biome is currently the main agricultural frontier of the country. Cerrado soils are mostly well drained and highly weathered, with a predominance of kaolinite clays [3] of low chemical reactivity and with limited carbon (C) [4]. It is imperative to preserve the soil quality of this important biome, which feeds a substantial part of the Brazilian and world population. Several products from the Cerrado are exported to various countries, e.g., soybean, maize, cotton, and cattle. In addition, the headwaters of the three rivers of São Francisco, Prata, and Araguaia/Tocantins, which form the main hydrographic basins in Latin America, lie in the Cerrado [5].

One of the key aspects for the evaluation of soil carbon stocks is the organic matter content, because a large part of the organic matter in the soil is constituted by organic carbon [6]. The preservation of or increase in C in soils is a challenge for Brazilian agriculture, particularly in the Cerrado, since the climatic conditions in this region favor rapid C losses in the soil. This factor not only reduces the soil fertility, but also affects the emission of greenhouse gases (GHG) into the atmosphere [7].

Several studies have already evaluated the C stocks in soils influenced by distinct agricultural practices applied in the Cerrado [8,9,10,11]. A recent review [12] observed a tendency of increases in annual C stock rates in Cerrado soils when climate-smart agriculture practices were adopted, consisting of soil fertilization with organic amendments, no-tillage cultivation, and crop–livestock or crop–livestock–forestry systems.

However, there is still no consensus in the literature about which of the agricultural practices contribute to increasing the soil C stocks in the Brazilian Cerrado. In many areas under systems of conservation agriculture, e.g., no-tillage with cover crops, the soil C stocks are lower than in adjacent areas under native Cerrado vegetation [8,13]; the same is true for areas under integrated cultivation systems [14,15]. On the other hand, studies have reported gains in soil C stocks after the introduction of agricultural practices, whereby the soil C stocks were higher than those under native Cerrado without anthropic interference [9,16,17].

In this review, we proposed an investigation of published papers in the literature, covering all agricultural practices used currently in the Brazilin Cerrado, to identify how agriculture has affected the C stocks of these soils and how it can affect GHG emissions into the atmosphere. The potential of C sequestration by soils is highly influenced by the initial soil C content [18]. Thus, this review took the values of the original C stocks in the soil into consideration, i.e., before the introduction of any agricultural practice, in comparison with the C stocks of the cultivated areas. 

At the beginning of the agricultural occupation of the Cerrado, the most used system was conventional cultivation, where the soil is ploughed, involving an intensive use of plow and harrow. Later, the no-tillage and integrated systems with crops, livestock, and forestry were introduced [19,20]. The data of this review represent these different agricultural practices.

Most studies have shown that C stocks in cultivated areas are lower than those found in native Cerrado soils (area with native vegetation, adjacent to the experimental sites). Nevertheless, when soil C stocks are assessed at greater depths (0–100 cm layer), 52.4% of the data from agricultural soils have higher values than in areas of the Cerrado without anthropogenic interference. This shows that it is possible to practice agriculture on the Cerrado soils that contribute to soil C sequestration. 

The questions of this review were: (1) Is it possible to have low-C agriculture on Brazilian Cerrado soils? (2) Can agriculture in the Cerrado increase soil C sequestration and, consequently, mitigate GHG emissions? (3) Does the sampling depth of C stocks interfere with the results? (4) Do the agricultural practices applied in the Brazilian Cerrado still need to be improved?

## 2. Results and Discussion

Figure 1 shows the geographic distribution of the studies across the Brazilian Cerrado. The studies were carried out in the different microregions of the Brazilian Cerrado and are therefore representative of the regionally used agricultural practices.

The selected studies deal with the different agricultural practices used in the Cerrado, namely, no-tillage systems, conventional tillage, monoculture, silvopastoral, integrated crop–livestock, and integrated crop–livestock–forest.

The articles selected for this review include several plant species currently cultivated in the soils of the Brazilian Cerrado (Table 1). The Cerrado vegetation sub-types considered in this review were Cerrado sensu stricto (shrubland, savanna-like vegetation with trees) and Cerradão (forest type savanna, closed canopy forest).

Most of the soil C stock data compiled in this review (185 of 283, or 65.37%) indicated that replacing the Cerrado vegetation with cultivated species reduced soil C stocks when compared with the initial soil C stocks, before the conversion to agriculture. On the other hand, 34.62% (98 of 283) of the soil C data indicated gains in the soil C stocks, i.e., it is possible to maintain the agricultural production on Cerrado soils and yet increase the soil C stocks. Depending on the agricultural practices, after the conversion of native Cerrado to agriculture, the soil can function a source or sink of atmospheric C [21]. Even under conservation managements, such as no-tillage or livestock–forestry integration, the original soil C stocks can be reduced [8,14,15].

Some studies in this review described agricultural practices by which the soil organic C stocks in Cerrado soil were increased by up to 93.6 Mg ha^−1^ [10]. However, other studies reported losses, which reached a maximum of 84.53 Mg ha^−1^ of C lost from the soil [8].

In agriculture, the applied practices affect the natural equilibrium of the soil and vegetation system. This is a consequence of soil tillage, fertilization, and the introduction of new plant species and pesticide application [22]. There is no way of practicing agriculture without any interference. The objective of this review is the identification of agricultural practices that cause the least possible impact, or which could even improve C stocks in the soils of the Brazilian Cerrado. The data of this review were grouped into the following layers: 0–30; 0–40; 0–60; and 0–100 cm. (Figure 2, Figure 3, Figure 4 and Figure 5).

### 2.1. Variation in the Soil Carbon Stocks at the 0–30 cm Depth 

For the 0–30 cm layer, 82 items of data from the studies in the articles of soil carbon stocks were analyzed (Figure 2). Of these, 26.8% (22 of 82) indicated increases after the implementation of agricultural practices, i.e., native Cerrado replacement by species of agricultural interest increased the soil C stocks. In addition, gains above 10 Mg ha^−1^ were recorded in five data items of soil C stocks [23,24]. Two of them were obtained under no-tillage cultivation and the other two under more than 20-year-old pastures.

**Figure 2 plants-12-02449-f002:**
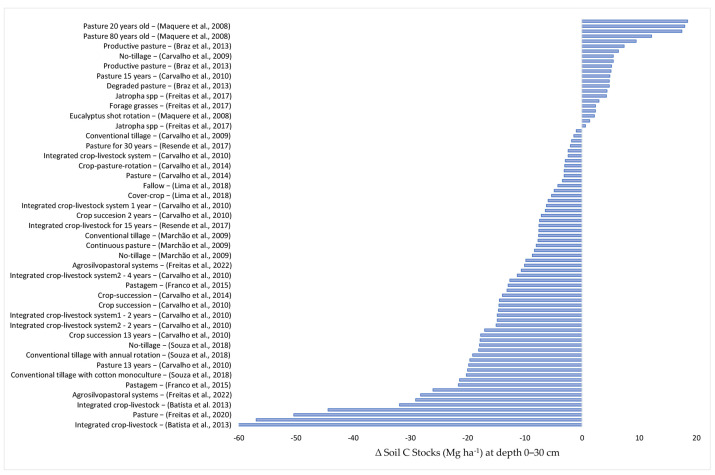
Soil carbon stock losses and gains (Mg ha^−1^), in response to different agricultural practices, in the 0–30 cm layer compared with the original C stocks in soil under Cerrado vegetation. The works cited are: (Batista et a., 2013) [14], (Freitas et al., 2020) [15], (Carneiro et al., 2013) [16], (Miranda et al., 2016) [17], (Rosinger et al., 2023) [18], (Costa et al., 2006) [19], (Silva et al., 2020) [20], (Carvalho et al., 2010) [21], (Dignac et al., 2017) [22], (Carvalho et al., 2009) [23], (Maquere et al., 2008) [24], (Freitas et al., 2022) [25], (Franco et al., 2015) [26], (Souza et al., 2018) [27], (Ferreira et al., 2016) [28], (Carvalho et al., 2014) [29], (Cruvinel et al., 2011) [30], (Marchão et al., 2009) [31], (Lima et al., 2018) [32], (Resende et al., 2017) [33], (Braz et al., 2013) [34], (Freitas et al., 2017) [35], (Carvalho et al., 2009) [23].

Nevertheless, losses of soil C stocks were also recorded in the 0–30 cm layer. Of the 82 data items of soil C stocks analyzed, 60 (73%) indicated a reduction in soil stocks compared to those in soils under native Cerrado (Figure 2). The highest reduction was 61.14 Mg ha^−1^, recorded in an Oxisol under Integrated Crop Livestock (ICL) cultivated with maize+brachiaria–cotton/soybean [14]. Rotations that include cotton can result in soil C losses due to the stalk management carried out in the areas with a tractor that moves the surface soil layer to control pests and diseases [21].

In the 0–30 cm layer, eight data items of soil C stock indicated losses above 25 Mg ha^−1^ [15,16,21,26]. Of these, three were determined in soils under integrated crop–livestock (ICL) (61.14 Mg ha^−1^, 31.89 Mg ha^−1^, 29.07 Mg ha^−1^), three in pasture soils (56.96 Mg ha^−1^, 50.38 Mg ha^−1^, 26.06 Mg ha^−1^), and two in soils under integrated crop–livestock–forestry (ICLF) (44.30 Mg ha^−1^ and 28.2 Mg ha^−1^). In general, the introduction of pastures, ICL, and ICLF increased the C stocks in the soil. Yet, when planting is carried out in areas with low C stocks, it takes a longer time until the increases in C stocks in these soils become perceptible [15].

### 2.2. Variation in the Soil Carbon Stocks at the 0–40 cm Depth 

Figure 3 lists 74 soil C stock data items of the 0–40 cm layer. Of these, 25 (33.7%) showed an increase in soil C stocks concerning the stocks recorded in the soil under Cerrado vegetation. Seven data items indicated gains above 10 Mg ha^−1^ [11,36,37,38]. Five of these represent soil under no-tillage cultivation and two soils under agroforestry. The highest gain in soil C stock was 59.63 Mg ha^−1^, in an Oxisol under agroforestry [11]. The diversification of plant species is one of the aspects that significantly contribute to increase soil C stocks [39]. This is not only due to the quantity, but, mainly, to the quality of these plant residues added to the soil with different C/N and lignin/N ratios [40], as is also the case in agroforestry systems [41].

**Figure 3 plants-12-02449-f003:**
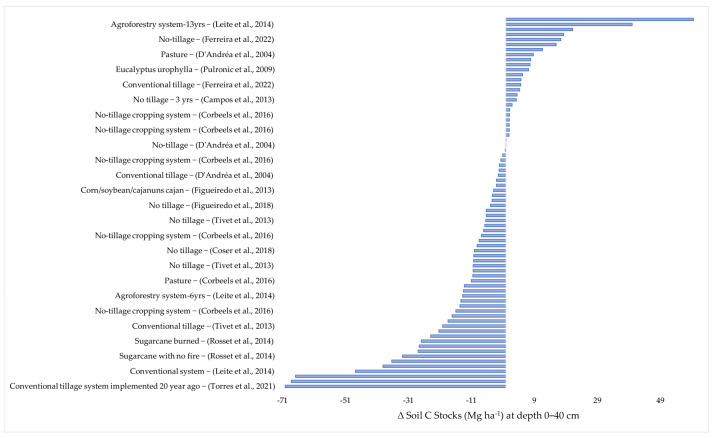
Soil carbon stock losses and gains (Mg ha^−1^), in response to different agricultural practices, in the 0–40 cm layer compared with the original C stocks in soil under Cerrado vegetation. The works cited are: (Rocha et al., 2014) [11], (Torres et al., 2021) [13], (Ferreira et al., 2022) [42], (Campos et al., 2013) [36], (Leite et al., 2014) [37], (Rosset et al., 2014) [38], (Torres et al., 2019) [43], (Coser et al., 2018) [40], (Corbeels et al., 2016) [41], (Tivet et al., 2013) [44], (Figueiredo et al., 2018) [45], (Figueiredo et al., 2013) [46], (Loss et al., 2012) [47], (Loss et al., 2013) [48], (D’Andréa et al., 2004) [49], (Vicentini et al., 2019) [50], (Pulrolnik et al., 2009) [51].

In the 0–40 cm layer, 49 (66.2%) of the data items indicated a reduction in the soil C stocks in relation to the native Cerrado areas. Ten data items of soil C stocks showed losses above 25 Mg ha^−1^, two of which represented crops under conventional tillage; four referred to crops under no-tillage and the other four were sugarcane monoculture [13,37,38,43]. The highest C losses were 70.13 Mg ha^−1^, recorded in an Oxisol cultivated for 20 years with maize/soybean/common bean/sorghum under conventional tillage [13].

### 2.3. Variation in the Soil Carbon Stocks at the 0–60 cm Depth

Figure 4 contains 10 data items of soil C stocks recorded in the 0–60 cm layer. Of these, three showed gains in the soil C stock compared to the native Cerrado [52,53]. Two of the data items referred to soils under no-tillage crops (soybean/*Brachiaria ruziziesis*/soybean and soybean/*B. ruziziensis*), and the gains were 5.2 Mg ha^−1^ and 4.4 Mg ha^−1^, respectively. The other C stock data were determined in soil under *Eucalyptus camaldulensis*, with a gain of 3.01 Mg ha^−1^. On the other hand, seven data items recorded decreases in soil C stocks, three of which exceeded reductions of 25 Mg ha^−1^ [54]. The highest loss in soil C stocks (50.1 Mg ha^−1^) was recorded in soil growing *Braquiaria brizantha* under conventional management [54], which shows that soil tillage accelerates organic matter decomposition and that a large part of the C is oxidized and returns to the atmosphere as CO_2_, reducing the capacity of the soil to accumulate/stock soil C.

**Figure 4 plants-12-02449-f004:**
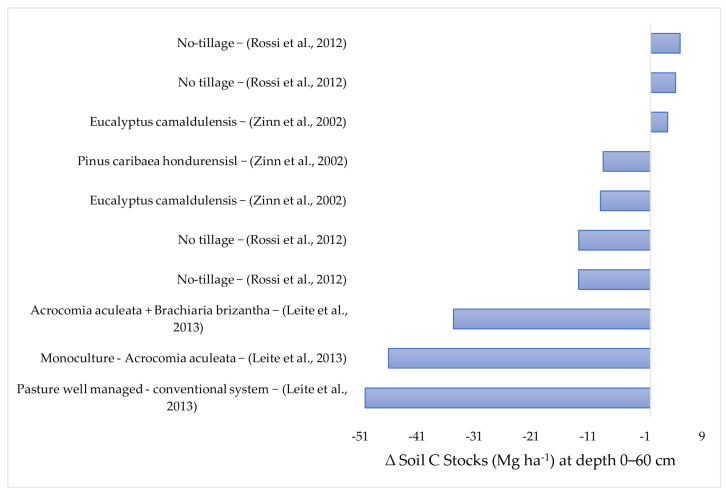
Soil carbon stock losses and gains (Mg ha^−1^), in response to different agricultural practices, in the 0–60 cm layer compared with the original C stocks in soil under Cerrado vegetation The works cited are: (Zinn et al., 2002) [52], (Rossi et al., 2012) [53], (Leite et al., 2013) [54].

### 2.4. Variation in the Soil Carbon Stocks at the 0–100 cm Depth 

Figure 5 shows 63 data items of soil C stocks recorded in the 0–100 cm layer. Of these, 33 (52.4%) indicate higher soil C stocks under agriculture than in Cerrado soils. The gains ranged from 1.3 Mg ha^−1^ to 93.6 Mg ha^−1^. This wide variation was due to differences in the systems, which included degraded to well-managed pastures. Twenty data items represented gains above 10 Mg ha^−1^ [10,17,23,34,51,55]. Of these, 14 were determined in Oxisol under pasture, three in soils under crop–livestock–forestry integration, two in no-tillage conditions, and one in soil under *Eucalyptus urophylla*. The gains in soil C stock were highest in Oxisol under *Urochloa decumbens* cv Basilisk and *Brachiaria brizantha* [10,34]. This can be explained by the high capacity of pastures to produce shoot and mainly root [56], and the positive effect of pasture management, by which a crop residue cover is left on the soil surface. Even in degraded areas, the biomass production by *Brachiaria* is significant [21]. Some *Brachiaria* species can adapt well to the soils of the Brazilian Cerrado, producing large quantities of the shoot and mainly root biomass and are efficient in storing C in the soil profile [56,57].

**Figure 5 plants-12-02449-f005:**
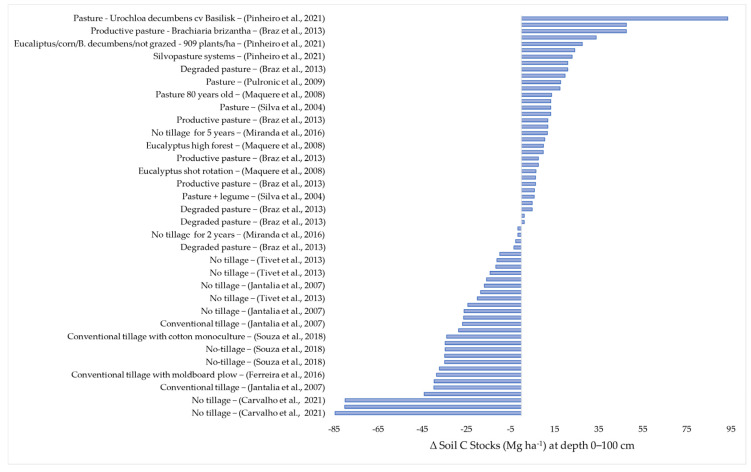
Soil carbon stock losses and gains (Mg ha^−1^), in response to different agricultural practices, in the 0−100 cm layer compared with the original C stocks in soil under Cerrado vegetation. The works cited are: (Carvalho et al., 2021) [8], (Pinheiro et al., 2021) [10], (Miranda et al., 2016) [17], (Maquere et al., 2008) [24], (Souza et al., 2018) [27], (Ferreria et al., 2016) [28], (Braz et al., 2013) [34], (Tivet al., 2013) [44], (Pulrolnik et al., 2009) [51], (Silva et al., 2004) [55], (Jantalia et al., 2007) [58].

Carbon losses were also recorded in the 0–100 cm layer. Thirty data items of soil C stocks indicated losses against those under native Cerrado. In 17 cases, the soil C stock data indicated losses above 25.0 Mg ha^−1^ [8,27,28,44,58]. Of these, nine were determined in soils under no-tillage cultivation and eight under conventional tillage. The highest losses (84.53 Mg ha^−1^) were recorded in soil with different cover crops under no-tillage cultivation [8]. 

To increase the soil C stocks, the agricultural practices must be adapted to local conditions and must have the capacity to increase biomass production and, consequently, the input of C into the soil. In [59], Rumpel et al. highlight that keeping the soil covered, as well as effective nutrient management, are effective agricultural practices to recover and increase carbon stocks in the soil. Simultaneously, they must prevent this C from being released into the atmosphere or transferred to other reservoirs at harvest [22]. In this sense, agricultural practices that favor the physical, physical–biochemical, and physical–chemical protection of organic carbon must be adopted in production systems. Good results were observed [60], with carbon accumulation distributed along the soil profile (0–100 cm) with manure application. 

The choice of plant species for cultivation is decisive for this purpose, since plants represent the link between C in the atmosphere and the C stored in the soil [61]. The diversification of plant species in agricultural systems with a grass–legume species mixture is recommended as a climate-smart farming practice [12,62]. Grasses can remove large amounts of C from the atmosphere, as they produce high biomass quantities [59]. Legumes will contribute to the nutrition of these plants, which can intensify biomass production without fertilizer application. To increase C stocks in Cerrado soils, [8,63] species such as *Urochloa ruziziensis* and *Canavalia brasiliensis* are recommended as cover crops due to the low C/N ratio, high hemicellulose, and low lignin contents.

The no-tillage system, without soil preparation, will contribute to a reduction in organic matter oxidation, with a consequent increase in soil C stocks. Conversely, the effect of the no-tillage approach may be low if the plants are not able to capture CO_2_ from the atmosphere [57]. Therefore, the choice of a diverse range of crop species for the production system is essential for higher soil C stocks. Gerke J [64] highlights that crop rotation with a mix of grasses and legumes, and species with great potential for biomass production associated with no tillage or minimum tillage is of fundamental importance in maintaining an increase in the soil organic carbon content.

The increase in soil C stocks in the Brazilian Cerrado depends on the species included in the cropping systems, and not only on not tilling the soil [61]. When species with a high C/N ratio are included, as in the case of grasses, it is more likely that C will be accumulated in the soils of this region. No-tillage systems with rapid plant decomposition do not contribute to C sequestration in soils in the Cerrado [31,57]. The introduction of legume species in pastures may favor an increase in soil C [65,66] as the nitrogen source can favor biomass production.

In the soil layers, we also observed that the effect of agricultural practices on the soil C stocks in the Brazilian Cerrado was more consistent when analyzing the 0–100 cm layer. Data from surface layers alone may not reflect the real changes that occurred as a result of agricultural practices. The effects of agricultural practices on soil C stocks may be imperceptible when only a depth of 0–30 cm is analyzed [17]. This can mainly be explained by the contribution of the root system of different plant species to increase soil C [17,61]. The carbon taken by the roots into the deeper layers of the soil has a great contribution to carbon sequestration, because in these layers, the carbon is protected against microbial attack due to the conditions of low oxygen concentration that reduce the microbial biomass as well as its activity [67].

### 2.5. Principal Component Analysis (PCA)

In order to study the relation between the edaphoclimatic factors that affect variations in the carbon stocks in each land use, a principal component analysis (PCA) was applied on a pair of matrices with two soil layers. These matrices were composed of 107 and 82 lines (carbon stock data) and 9 columns (categorical and response variables), respectively, for the most relevant soil layers (0–30 and 0–100 cm). Soil type and land use were used as categorical variables and clay content, time after conversion, soil carbon stocks, delta C stocks, mean temperature, and rainfall as response variables.

Table 2 shows the results for the principal component analysis for two depths. For the 0–30 cm layer (Table 2, Figure 6a), the PCA indicates that 57.3% of the original data could be explained by the two main components. For the 0–100 cm layer (Table 2, Figure 6b), the PCA indicates that 59.2% of the original data could be explained by the two main components. It was observed that, in the 0–30 cm layer, the soil C stock and clay content were strongly correlated. These variables showed the higher values of loading at the PC1 axis. At the PC 2 axis, the higher value of loading was related to the delta soil C stock. For the 0–100 cm layer, the soil C stock and the clay content were strongly correlated. These variables showed the higher values of loading at the PC1 axis. At the PC 2 axis, the higher value of loading was related to the delta soil C stock.

## 3. Materials and Methods

This systematic review was carried out according to a protocol proposed by [68], based on the PRISMA flowchart. The first step of this review was the definition of the research questions. 

The research protocol was based on a combination of keywords. The terms were searched for in the title, abstract, and keywords of published papers. We used the Scopus database to collect the publications.

To answer the research questions, we selected and clustered the keywords as follows: (i) (“integrated crop-livestock” AND “soil carbon stock” AND “cerrado” OR “savanna” AND Brazil), with 215 results; (ii) (“no-tillage” AND” soil carbon stock” AND “cerrado” OR “savanna” AND Brazil), with 216 results; (iii) (“cerrado” AND “carbon stock” AND “soils”), with 99 results; (iv) (“cerrado”, “carbon stock”, “soils” AND “organic matter”), with 51 results; (v) (“soil carbon” OR “soil organic carbon” OR “soil organic matter”) AND (“cerrado” OR “brazilian savanna”) AND (“greenhouse gas”), with 30 results; (vi) (((“organic matter” AND “greenhouse gas”) AND (“cerrado” OR “brazilian savanna”))), with 17 results; (vii) (((“carbon stock” AND “greenhouse gas”) AND (“cerrado” OR “brazilian savanna”))), with 18 results; (viii) (((“organic matter” AND “root pasture”) AND (“cerrado” OR “brazilian savanna”))), with 18 results. Based on the examination of these eight conditions, a total of 664 articles (only complete articles) were found.

After eliminating duplicates, 475 articles remained. The abstracts of these 475 were read to select those in which field experiments were carried out in the Cerrado, with measurements of C stocks in the soil profile. To this end, we used the following exclusion criteria:

Exclusion Criterion 1—Review articles, as we were looking for original soil C stock data;

Exclusion Criterion 2—Carbon modeling papers to estimate soil C stocks;

Exclusion criterion 3—Articles that did not use an area of native Cerrado vegetation adjacent to the experimental sites as a reference for the comparison of C stocks;

Exclusion criterion 4—Articles in which the data were expressed in figures and the exact value of soil C stocks could not be identified.

After applying these four exclusion criteria, 63 articles remained for data extraction. The data were extracted from tables and figures containing the values of C stocks in Mg ha^−1^, measured in soils under agricultural areas and of the reference area—in this case, Cerrado soil without anthropogenic influence.

A spreadsheet was set up with all of the values of the C stocks of the cultivated and native vegetation areas. Then, the differences between the original C stocks in the soil under native Cerrado and agricultural areas were calculated. These differences were expressed in Mg ha^−1^ of C. Subsequently, the data were grouped according to the soil layers: 0–30, 0–40, 0–60, and 0–100 cm. We did not group the C stock data of layers thinner than 30 cm. Next, the data were organized in ascending order. A graph was created for each layer (see Results and Discussion).

Principal component analysis (PCA) was applied to the database extracted from the articles. The quantitative variables selected were mean temperature (°C), average annual rainfall (mm), time since land use change (years), clay content (g kg^−1^), soil C stocks (Mg ha^−1^), and ∆ C stocks (Mg ha^−1^), calculated as the difference between the soil C stocks under agricultural areas and those under native Cerrado as reference. The soil sampling layers (0–30 and 0–100 cm) corresponded to the qualitative variable analyzed and were defined as recommended by the Intergovernmental Panel on Climate Change (IPCC) (0–30 cm) and the deepest profile usually sampled (0–100 cm).

Before the PCA, measures of sampling adequacy were calculated by the Kaiser–Meyer–Olkin (KMO) and Bartlett’s sphericity tests (*p* < 0.05). Values of more than 0.5 to 1.0 for KMO are considered acceptable for PCA application [69]. The independence of the variables was checked by Bartlett’s sphericity test. Principal component analysis was performed to group the dataset into new variables that resume the information in principal components (PC). The analysis also prevented multicollinearity between the original variables. The PC had the objective of explaining as much of the variation in the original variables as possible.

## 4. Conclusions

The contribution of this study is that it shows that the agricultural practices implemented in the Brazilian Cerrado make low-carbon agriculture in this biome possible. In addition to agricultural production, farming in the Brazilian Cerrado can also contribute to soil carbon sequestration and, consequently, the mitigation of greenhouse gas emissions. We presented some studies with gains in soil carbon stocks after the implementation of agricultural management. 

In addition, the effect of the agricultural management on soil carbon stocks in the Brazilian Cerrado was more consistent when the 0–100 cm layer was taken into consideration. 

However, the agricultural practices implemented in the Brazilian Cerrado can be further improved to favor and recover the lost soil carbon, and even increase its levels. Although soil C storage capacity is low due to the low clay reactivity, the original soil carbon stocks could not be recovered so far by the agricultural managements applied. For the future, we suggest new research in search of agricultural practices that can increase soil carbon stocks and mitigate GHG in the Brazilian Cerrado, with a special emphasis on plant species diversity in the production systems and cover crops with a high C/N ratio in legume–grass mixtures. The evaluations should take a soil layer of at least 0–100 cm into consideration.

## Figures and Tables

**Figure 1 plants-12-02449-f001:**
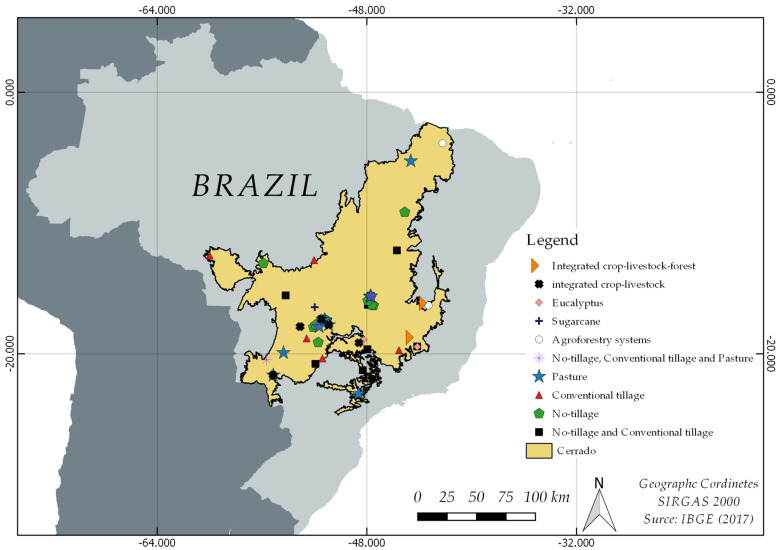
Distribution of locations of experimental studies analyzed in this review, in the Brazilian Cerrado.

**Figure 6 plants-12-02449-f006:**
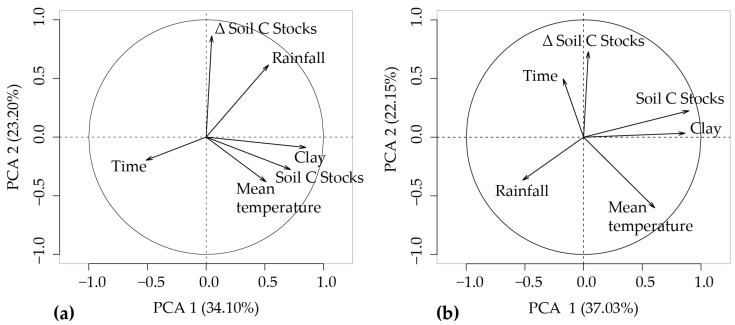
Principal components for (**a**) 0–30 cm and (**b**) 0–100 cm layers.

**Table 1 plants-12-02449-t001:** Cultivated species found in the database obtained for the Cerrado with disponible soil carbon stock data.

Scientific Name	Common Name
**Crops**	
*Glycine max*	Soybean
*Zea mays*	Maize
*Sorghum bicolor*	Sorghum
*Oryza sativa*	Rice
*Phaseolus vulgaris*	Common bean
*Solanum lycopersicum*	Tomato
*Gossypium hirsutum*	Cotton
*Jatropha* spp.	Jatropha
*Saccharum officinarum*	Sugarcane
**Cover crops**	
*Brassica rapa*	Turnip
*Gliricidia sepium*	Gliricidia
*Crotalaria* sp	Crotalaria
*Cajanus cajan*	Pigeon pea
*Canavalia ensiformis*	Pork bean
*Pennisetum glaucum*	Millet
**Pastures**	
*Brachiaria decumbens*	Brachiaria grass
*Brachiaria brizantha*	Palisade grass
*Panicum maximum* cv. Mombaça	Mombaça grass
*Panicum maximum* cv. Tanzânia	Tanzania grass
*Cynodon* spp.	Bermuda grass
*Cenchrus ciliaris*	Buffalo grass
*Cenchrus echinatus*	Burr grass
*Paspalum atratum*	Pojuca Grass
**Forestry**	
*Hevea brasiliensis*	Rubber tree
*Eucalyptus urophylla*	Eucalyptus urophylla
*Eucalyptus grandis*	Rose gum
*Pinus caribaea hondurensis*	Pinus
*Acrocomia aculeata*	Coconut palm

**Table 2 plants-12-02449-t002:** Summary of principal component analysis (PCA) for the 0–30 and 0–100 cm layers.

	PC1	PC2	PC1	PC2
Depth	0–30 cm		0–100 cm	
Variability (%)	34.1	23.2	37.0	22.2
Cumulative (%)	34.1	57.3	37.0	59.2
Clay	0.85	−0.089	0.864	0.034
Time	−0.513	−0.197	−0.174	0.497
Soil carbon stocks	0.719	−0.277	0.898	0.226
Δ Soil carbon stocks	0.040	0.863	0.04	0.729
Mean temperature	0.51	−0.382	0.606	−0.603
Rainfall	0.531	0.615	−0.519	−0.367

## Data Availability

The data presented in this study are available on request from the corresponding author.

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
