# Peer review of "Soil Carbon Stocks and Greenhouse Gas Mitigation of Agriculture in the Brazilian Cerrado—A Review"

_plants, 2023, doi:10.3390/plants12132449_

Round 1
Reviewer 1 Report
It is important to evaluate soil carbon dynamics in agricultural ecosystems in light of sustainable management and CO2 mitigation. The study reviewed soil carbon change by comparing different agricultural practices with native vegetation. Results presented in this study bring a benchmark for carbon counting in agricultural ecosystems and provide possible low-carbon agricultural practices in the Brazilian Cerrado.
The study method is sound and results are well clearly presented.
I have two minor suggestions for the authors.
1. Denote different agricultural practices with different legends in Figure 1.
2. Focus on topsoil (0-30 cm) and whole soil profile (0-100 cm)is enough when evaluating soil carbon stock variation under different practices.
Author Response
Dear Prof. Dr. Dilantha Fernando,
We thank the reviewers for their generous and thoughtful comments on our submitted manuscript. We have made all the efforts to address all the reviewers’ concerns. We have included here the reviewers’ comments and how we have addressed each point, in italics the reviewers’ comments and in color blue our response.
- -Reviewer 1 –
“… Denote different agricultural practices with different legends in Figure 1.
Thank you for this suggestion. We have modified Figure 1 including the different agricultural practices and respective legend.
- -Reviewer 1 –
“… Focus on topsoil (0-30 cm) and whole soil profile (0-100 cm)is enough when evaluating soil carbon stock variation under different practices.
Thank you for this comment. We definitely agree with your comment, and our text discusses and concludes on a similar line. Because this is a review we brought considerations from the studies in the articles as well, but the PCA analysis also included enhances those lines.
“The soil sampling layers (0-30 and 0-100 cm) corresponded to the qualitative variable analyzed and were defined as recommended by the Intergovernmental Panel on Climate Change (IPCC) (0-30 cm) and the deepest profile usually sampled (0-100 cm).”
We would be glad to respond to any further questions and comments that you may have.
On behalf of all authors. Sincerely yours,

Reviewer 2 Report
The present manuscript entitled "Soil carbon stocks and greenhouse gas mitigation of agriculture in the Brazilian Cerrado – a review is an interesting work aimed at the identification of agricultural practices that cause the least possible impact, or which could even improve C stocks in the soils of the Brazilian Cerrado. Based on my review, I suggest a minor revision. My detailed and specific comments can be found in the attached PDF.

I have found some minor English language mistakes and typos which I have highlighted in the PDF attached.
Author Response
Dear Prof. Dr. Dilantha Fernando,
We thank the reviewers for their generous and thoughtful comments on our submitted manuscript. We have made all the efforts to address all the reviewers’ concerns. We have included here the reviewers’ comments and how we have addressed each point, in italics the reviewers’ comments and in color blue our response.
- -Reviewer 2 –
“… Please refer to the following articles:
Rumpel, C., Amiraslani, F., Bossio, D., Chenu, C., Cardenas, M.G., Henry, B., Espinoza, A.F., Koutika, L.S., Ladha, J., Madari, B.E. and Minasny, B., 2023. Studies from global regions indicate promising avenues for maintaining and increasing soil organic carbon stocks: The Scientific and Technical Committee of the 4 per 1000 initiative. Regional Environmental Change, 23(1), p.8.
Abrar, M.M., Xu, M., Shah, S.A.A., Aslam, M.W., Aziz, T., Mustafa, A., Ashraf, M.N., Zhou, B. and Ma, X., 2020. Variations in the profile distribution and protection mechanisms of organic carbon under long-term fertilization in a Chinese Mollisol. Science of the Total Environment, 723, p.138181. …
Thank you very much for these comments. We have included the mentioned articles (lines 226—233) in the Results and Discussion section, and referred to them in the bibliography.
Lines 226—233
In [66], Rumpel et al. highlight that keeping the soil covered, as well as effective nutrients management, are effective agricultural practices to recover and increase carbon stocks in the soil. Simultaneously, they must prevent this C from being released into the atmosphere or transferred to other reservoirs at harvest [23]. In this sense, agricultural practices that favor the physical, physical-biochemical, and physical-chemical protection of organic carbon must be adopted in production systems. Was observed good results [67], with carbon accumulation distributed along the soil profile (0-100 cm), with manure application.
- -Reviewer 2 –
“… typos and English suggestions marked on the text:
Thank you very much for the corrections and suggestions on the English text. We have corrected and revised all the text following your remarks.
- -Reviewer 2 –
“… Please also provide the common name of the crops.
I suggest the authors to give the information in a table.:
Thank you very much for suggesting those. We have included all the common names of the crops in a new table (Table 1) as requested.
- -Reviewer 2 –
“… These are general C stocks or the organic C stocks? (line 122).
Thank you very much the question. The answer is soil organic C stocks, and we have also clarified that in the text (line 121).
- -Reviewer 2 –
“… Do you mean 82 articles? Please clarify throughout the text. (line 134).
Thank you very much the question. No, there are 82 data from the studies realized in the articles, since some articles show more than one study. We have modified the phrase (line 134) to make it clearer.
We would be glad to respond to any further questions and comments that you may have.
On behalf of all authors. Sincerely yours,
